# Biostimulation of Microbial Communities from Malaysian Agricultural Soil for Detoxification of Metanil Yellow Dye; a Response Surface Methodological Approach

**Fatin Natasha Amira Muliadi [1], Mohd Izuan Effendi Halmi [1],\*, Samsuri Bin Abdul Wahid [1], Siti Salwa Abd Gani [2], Uswatun Hasanah Zaidan [3], Khairil Mahmud [4] and Mohd Yunus Abd Shukor [3]**

1. Department of Land Management, Faculty of Agriculture, Universiti Putra Malaysia, Serdang 43400, Selangor, Malaysia; fatin.muliadi@gmail.com (F.N.A.M.); samsuriaw@upm.edu.my (S.B.A.W.)
2. Department of Agricultural Technology, Faculty of Agriculture, Universiti Putra Malaysia, Serdang 43400, Selangor, Malaysia; ssalwaag@upm.edu.my
3. Department of Biochemistry, Faculty of Biotechnology and Biomolecular Sciences, Universiti Putra Malaysia, Serdang 43400, Selangor, Malaysia; uswatun@upm.edu.my (U.H.Z.); mohdyunus@upm.edu.my (M.Y.A.S.)
4. Department of Crop Science, Faculty of Agriculture, Universiti Putra Malaysia, Serdang 43400, Selangor, Malaysia; khairilmahmud@upm.edu.my
\* Correspondence: m_izuaneffendi@upm.edu.my; Tel.: +60-3-97694958

**Abstract:** In the present study, a mixed culture from a local agricultural soil sample was isolated for Metanil Yellow (MY) dye decolorization. The metagenomic analysis confirmed that 42.6% has been dominated by genus *Bacillus*, while *Acinetobacter* (14.0%) is present in the microbial communities of the mixed culture. For fungi diversity analysis, around 97.0% was "unclassified" fungi and 3% was Candida. The preliminary investigation in minimal salt media (MSM) showed that 100% decolorization was achieved after 24 h of incubation. Response surface methodology (RSM) was successfully applied using Box-Behnken design (BBD) to study the effect of four independent parameters—MY dye concentration, glucose concentration, ammonium sulfate concentration, and pH—on MY dye decolorization by the mixed bacterial culture. The optimal conditions predicted by the desirability function were 73 mg/L of MY, 1.934% glucose, 0.433 g/L of ammonium sulfate, and a pH of 7.097, with 97.551% decolorization The correlation coefficients ($R^2$ and $R^2$ adj) of 0.913 and 0.825 indicate that the established model is suitable to predict the effectiveness of dye decolorization under the investigated condition. The MY decolorization of the mixed bacterial culture was not affected by the addition of heavy metals in the growth media. Among the 10 heavy metals tested, only copper gave 56.19% MY decolorization, whereas the others gave almost 100% decolorization. The decolorization potential of the mixed bacterial culture indicates that it could be effective for future bioremediation of soil-contaminated sites and treatment solutions of water bodies polluted with the MY dye.

**Keywords:** bioremediation; dye decolorization; mixed culture; metanil yellow; response surface methodology

## 1. Introduction

Azo dyes are widely used in various industries, such as textile, food, paper-making, and cosmetic industries [1]. Azo dyes constitute about one million tons of production, and about 300,000 tons of different dyestuffs have been used per year for the operations of textile dyeing [2]. The increased demand for azo dye textile products has produced effluents that lead to severe water pollution. Metanil Yellow (MY) (monosodium salt of 4-m-sulphophenylazodyephenylamine) is one type of acidic azo dye. MY is commonly used as soap coloring, spirit lacquer, shoe polish, and bloom sheep dip, as well as for the preparation of food stains, leather dyeing, the manufacturing of pigment lakes, and paper staining. It is also used as food colorants in various foodstuffs, mostly in India [2]. However, it was later discovered that MY is carcinogenic [2]. From previous toxicity data, it was shown

that when MY was fed to animals, the animals developed testicular lesions, leading to a decrease in the rate of spermatogenesis [2]. According to Saratale (2009), exposure to the dyes leads to potential health hazards, such as asthma, rhinitis, and dermatitis. Apart from the hazards to human health, azo dyes also affect ecosystems. When wastewater effluents are discharged into open water sources, they affect the photosynthetic activity of aquatic organisms [3], and the dissolved dyes may also affect the aquatic organisms since the breakdown products may also be toxic [4].

Removal of dyes from wastewater has gained scientific interest among researchers. As they are relatively recalcitrant to biodegradation, the elimination of colored effluents in wastewater treatment systems is mainly based on physical or chemical procedures, such as adsorption, chemical transformation, and incineration [5]. However, these methods are very costly, have high sludge production, and result in the formation of secondary by-products [6]. Therefore, biological methods for removing dyes must be developed because they are environmentally friendly, economical, and more cost-effective compared to physical or chemical procedures [7]. Biodegradation by different microorganisms, which uses bacteria as a dye decolorizing agent, appears to be an attractive alternative [8].

Microbial decolorization occurs under several conditions, such as anaerobic, anoxic, and aerobic conditions. For decolorization under aerobic conditions, the bacteria usually require organic carbon sources since they cannot utilize the dye as a growth substrate [8]. Generally, the degradation and decolorization of azo dyes by bacteria proceeds in two stages. The first stage involves reductive cleavage of the dyes' azo linkages, resulting in the formation of generally colorless but potentially hazardous aromatic amines. The second stage involves the degradation of aromatic amines [9]. Reductive cleavage of the –N=N– bond is the initial step in the bacterial degradation of azo dyes. It can occur under different types of mechanisms, such as through enzymatic reactions involving azoreductase and laccase enzymes, low-molecular-weight redox mediators, chemical reduction by biogenic reductants like sulphide, or a combination of these mechanisms [8]. The use of enzymes is beneficial due to their substrate specificity and may be effectively used in textile water pretreatment [10]. During azoreductase, enzymatic dye degradation, the azo bond (–N=N–) is cleaved by the enzyme and four electrons are transferred as the reducing equivalent. In each stage, two electrons are transferred to the azo dye, which is the electron acceptor, and decolorization occurs when the colorless solution is formed. The resulting intermediate is a toxic aromatic amine, which is later degraded by the aerobic process or sometimes microaerophilically [10].

It has been reported that bacterial dye decolorization and degradation can occur by a pure culture of bacteria and by a mixed culture of bacteria. According to a previous study, it has been reported that mixed bacterial culture can give a better degradation rate than the individual strain [11]. Individual species have the limited metabolic capability to mineralize the dye completely, and in many cases, it has been clearly observed that due to the lack of a catabolic pathway, aromatic amines are not further degraded. Catabolic and syntrophic interactions between indigenous species lead to complete degradation of azo dyes [12]. Azo dyes are not readily metabolized under aerobic conditions, and as a result of metabolic pathways, they degrade into intermediate compounds but not mineralized. They can be completely degraded under coupled aerobic and anaerobic degradation. Therefore, coupled anaerobic treatment followed by aerobic treatment can be an efficient and effective degradation method of azo dyes.

The One Factor at a Time (OFAT) method involves varying a single independent variable while the other variables are constant. This approach, however, is laborious, time-consuming, and incomplete. Thus, response surface methodology (RSM) using Box–Behnken design (BBD), which involves examining the simultaneous, systematic, and efficient variation of the important parameters used to model the decolorization process, identifies the possible interactions and higher-order effects and determines the optimum operational conditions [13].

The Box–Behnken statistical structure is an RSM plan with an autonomous, rotatable quadratic plan with treatment blends at the inside and midpoints at the edges of the procedure space. This structure requires less exploratory runs and less time compared to other RSM models. Thus, it is a more practical system. RSM is a collection of statistical and mathematical techniques that are useful for developing, improving, and optimizing processes. In this study, decolorization of MY by a newly isolated mixed bacterial culture from agricultural soil was optimized using RSM. The BBD matrix was chosen, and MY dye concentration, glucose concentration, ammonium sulfate concentration, and pH were used in this optimization study. This study was carried out to isolate, characterize, and optimize MY decolorization from various agricultural soil samples. It was discovered that most of the bacterial degrading dye was isolated from non-agricultural soil. Thus, the isolation of mixed bacterial cultures from agricultural soil samples improves the bioremediation process. To the best of our knowledge, no study has reported the application of RSM for optimizing the media used in the decolorization of the MY dye. Thus, this finding will provide significant results for MY decolorization using mixed bacterial culture.

## 2. Materials and Methods

### 2.1. Isolation and Screening of Mixed Cultures of Bacteria from Agricultural Soil

Forty samples of agricultural soils from different locations were collected for the isolation of bacteria. All soil samples were screened to isolate and identify the presence of the MY decolorizing mixed bacterial culture from agricultural soil and non-dye-contaminated environments. Until today, no studies have shown that mixed bacterial cultures isolated from non-dye-contaminated environments have been able to decolorize MY. Thus, this demonstrates the novelty of this research. For the first screening, 1 g of soil was weighed and grown in minimal salt media (MSM) composed of glucose (1%), $(NH_4)_2SO_4$ (0.40 g/L), $KH_2PO_4$ (0.20 g/L), $K_2HPO_4$ (0.40 g/L), NaCl (0.10 g/L), $Na_2M_oO_4$ (0.01 g/L), $MgSO_4.7H_2O$ (0.10 g/L), $Fe_2(SO_4)_3H_2O$ (0.01 g/L), $MnSO_4.H_2O$ (0.01 g/L), and yeast extract (1.00 g/L) supplemented with 50 mg/L of MY dye in a 250 mL conical flask. The cultivation of bacteria was conducted at room temperature on a rotary shaker (120 rpm). After the first screening, the secondary screening was performed by exposing the chosen mixed culture from the first screening to different dye concentrations ranging from 100 mg/L to 400 mg/L. The absorbance of the dye was measured by aseptically drawing 1.0 mL of the growth media and centrifuged it at $10,000 \times g$ for 10 min. The absorbance of the supernatant of the sample was measured at 434 using a UV—visible spectrophotometer. The mixed bacterial culture that was able to decolorize a high dye concentration was chosen for further study. The chosen mixed bacterial culture was maintained by subculturing in MSM consisting of 100 mg/L of the MY dye and 1.0% glucose in a conical flask. The decolorizing efficiency of the dye was expressed as the percentage of decolorization.

$$\% \text{ Decolourisation } = \frac{(\text{Initial absorbance} - \text{Final absorbance})}{\text{Initial absorbance}} \times 100\%.$$

### 2.2. Identification of the Chosen Mixed Bacterial Culture Using Metagenomic Analysis

The metagenomic analysis is a method that is used to investigate complex microbial communities from environmental samples without culturing or isolating a single organism [14]. It is estimated that about 99% of microorganisms that are present in natural environments are not readily culturable, and this has made it impossible to investigate the functional roles of different microbes in a certain niche. Therefore, metagenomics studies have made it possible to analyze complex genomes in a microbial community [15].

In this study, the chosen mixed bacterial culture was sent to Apical Scientific Sdn Bhd for metagenomics analysis. 16s rRNA is used for the diversity analysis of bacteria communities, while the internal transcribed spacer (ITS) is used for the diversity analysis of fungi. To identify the genus of a certain microorganism, the 16s rRNA gene is used since it is the "gold standard" that is routinely used in classifying prokaryotes. For the metagenomic

analysis, the steps include experimental design, sampling, sample fractionation, DNA extraction, DNA sequencing, assembly, binning, annotation, and statistical analysis [16,17].

In brief, the forward and reverse reads were merged using FLASH 2 and quality screened for sequence length and nucleotide ambiguity. All sequences that were shorter than 150 bp or longer than 600 bp (sequenced on the MiSeq platform) were removed from downstream processing. Reads were then aligned with 16S rRNA or the UNITE ITS database and inspected for chimeric errors. After these quality assessment steps, reads were clustered at 97% similarity into operational taxonomy units (OTUs).

*2.3. Optimisation of Significant Parameters Using Response Surface Methodology (RSM)*

The experimental process of optimizing MY dye decolorization was conducted using RSM with the BBD as an experimental matrix. In this study, RSM was used to investigate the relationships between four different significant parameters—dye concentration (A), glucose concentration (B), ammonium sulfate concentration (C), and pH (D)—and dye decolorization as a response and to optimize the relevant conditions of the variables and to predict the optimal conditions. The results obtained from the experiment were statistically analyzed using the Design Expert version 6.0.10, and a total of 29 runs were carried out. The results were analyzed using the Design Expert add-ons program, including analysis of variance (ANOVA), to determine the interaction between the variables and the response. The quality of the fit of this model was expressed by the coefficient of determination ($R^2$) in the same program [18]. All of the parameters were studied based on the range in Table 1.

**Table 1.** The upper limit and lower limit of Box–Behnken design (BBD).

| Parameters | Unit | Upper Limit | Lower Limit |
|---|---|---|---|
| Dye concentration | mg/L | 200.0 | 50.0 |
| Glucose concentration | % | 2.0 | 0.5 |
| Ammonium sulphate concentration | g/L | 1.0 | 0.1 |
| pH | | 6.0 | 7.5 |

To start the experiment, 1.0% of the fresh mixed bacterial culture was grown in MSM supplemented with various concentrations of dye (A), glucose (B), and ammonium sulfate (C) and at varying pH values (D), based on Table 2. The design experiments were carried out in conical flasks and the conical flasks were incubated on a rotary shaker (120 rpm) for 24 h. After 24 h, 1.0 mL of the sample was withdrawn and centrifuged at 10,000× *g* for 10 min. The supernatant of the sample was measured at 434 nm using a UV–visible spectrophotometer, and the percentage of dye decolorization was calculated using the formula given above.

*2.4. Effect of Different Initial Dye Concentrations on Dye Decolorisation and Bacterial Growth*

The mixed bacterial culture was grown in MSM supplemented with 50 mg/L of MY and incubated on a rotary shaker for 24 h. Next, 1.0% of the mixed bacterial culture was inoculated in MSM supplemented with different concentrations of MY (30, 60, 90, 120, 130, 150, 180, 210, 240, 270, and 300 mg/L). This was performed in duplicates. They were then incubated at 120 rpm on a rotary shaker at room temperature. Every 2 h for the first 24 h, 1.0 mL of the culture was aseptically withdrawn and centrifuged at 10,000× *g* for 10 min. The supernatant was measured at 434 nm. After 24 h, the reading was taken every 6 h until 66 h, and then the reading was taken every 12 h until 114 h.

*2.5. Effect of Heavy Metal Ions on Dye Decolorisation*

In this study, the effect of heavy metals on the dye colorization of free mixed bacterial was evaluated. MSM was separately supplied with 1.0 mg/L of copper (Cu), arsenic (As), zinc (Zn), chromium (Cr), nickel (Ni), silver (Ag), lead (Pb), and mercury (Hg). The free mixed bacterial cells were prepared based on optimum conditions that were previously obtained from RSM. The mixed bacterial culture was grown in MSM supplemented with

50 mg/L of MY dye and 1.0 mg/L of the heavy metals listed. This was performed in duplicates. The mixed bacterial culture was then incubated at 120 rpm on a rotary shaker at room temperature. Bacterial cultures without metals served as controls. After 24 h of incubation, 1.0 mL of the culture was withdrawn aseptically and centrifuged at $10,000 \times g$ for 10 min. The supernatant was measured at 434 nm, and the percentage of dye decolorization was calculated using the formula above.

**Table 2.** Experimental runs predicted by response surface methodology (RSM).

| Run | Factor 1 A: Dye Concentration (mg/L) | Factor 2: B: Glucose Concentration (%) | Factor 3: C: Ammonium Sulphate Concentration (g/L) | Factor 4: D: pH |
|-----|------|------|------|------|
| 1 | 50 | 2.00 | 0.55 | 6.75 |
| 2 | 50 | 1.25 | 0.55 | 7.50 |
| 3 | 125 | 1.25 | 1.00 | 6.00 |
| 4 | 125 | 1.25 | 1.00 | 7.50 |
| 5 | 200 | 1.25 | 0.55 | 7.50 |
| 6 | 200 | 1.25 | 1.00 | 6.75 |
| 7 | 200 | 1.25 | 0.55 | 6.00 |
| 8 | 125 | 2.00 | 1.00 | 6.75 |
| 9 | 50 | 1.25 | 0.55 | 6.00 |
| 10 | 50 | 0.50 | 0.55 | 6.75 |
| 11 | 125 | 200 | 0.55 | 6.00 |
| 12 | 125 | 1.25 | 0.10 | 6.00 |
| 13 | 200 | 1.25 | 0.10 | 6.75 |
| 14 | 125 | 0.50 | 1.00 | 6.75 |
| 15 | 125 | 0.50 | 0.55 | 6.00 |
| 16 | 125 | 0.50 | 0.10 | 6.75 |
| 17 | 125 | 1.25 | 0.55 | 6.75 |
| 18 | 50 | 1.25 | 1.00 | 6.75 |
| 19 | 200 | 0.50 | 0.55 | 6.75 |
| 20 | 125 | 1.25 | 0.55 | 6.75 |
| 21 | 50 | 1.25 | 0.10 | 6.75 |
| 22 | 125 | 2.00 | 0.10 | 6.75 |
| 23 | 125 | 2.00 | 0.55 | 7.50 |
| 24 | 200 | 2.00 | 0.55 | 6.75 |
| 25 | 125 | 1.25 | 0.55 | 6.75 |
| 26 | 125 | 1.25 | 0.55 | 6.75 |
| 27 | 125 | 1.25 | 0.55 | 6.75 |
| 28 | 125 | 1.25 | 0.10 | 7.50 |
| 29 | 125 | 0.50 | 0.55 | 7.50 |

## 3. Results

### 3.1. Isolation and Screening of Mixed Cultures of Bacteria from the Soil Sample

In this study, 40 agricultural soil samples were collected for the mixed culture MY degrader. After the secondary screening process, isolates, namely FN3 from oil palm estate soil collected from the Universiti Putra Malaysia (UPM), GPS 2.9876, 101.7234, were selected for further study. This was because isolate FN3 was able to decolorize up to 200 mg/L of dye compared to the other isolates, which could only decolorize up to 100 mg/L of MY dye. Isolate FN3 was able to decolorize 92.93% of 200 mg/L of MY, whereas the other isolates were only able to decolorize 100 mg/L of MY [19].

### 3.2. Metagenomic Analysis of Mixed Bacterial Culture

We performed a metagenomics analysis of mixed bacterial culture FN3 by using 16s rRNA for bacterial identification and ITS for fungi identification. Bacterial 16s rRNA genes have nine hypervariable regions (V1–V9) that indicate sequence diversity among the different bacteria. This is used for bacterial diversity analysis [14,20]. For fungi diversity

analysis, ITS was used. ITS is located between the 18S and 5.8S rRNA genes and has a high degree of sequence variation. The chosen mixed bacterial culture was sent to Apical Scientific Sdn. Bhd, Malaysia for metagenomic analysis. For bacterial identification using 16S rRNA, out of the nine hypervariable regions, regions V3–V4 were chosen as the primer. The sequences were grouped and clustered into OTUs. In brief, the forward and reverse reads were merged using FLASH 2 and quality screened for sequence length and nucleotide ambiguity. All sequences that were shorter than 150 bp or longer than 600 bp (sequenced on the MiSeq platform) were removed from downstream processing. Reads were then aligned with the 16S rRNA database and inspected for chimeric errors. After these quality assessment steps, reads were clustered at 97% similarity into OTUs; rare OTUs with only one (singleton) or two reads (doubleton), which are often spurious, were deleted from downstream processing.

We identified 97% as Kingdom bacteria which is a major part of the microbial communities. Out of this 97%, 45% was comprised of phylum Firmicutes, 44% was comprised of phylum Proteobacteria, and 9% was comprised of phylum Bacteroidetes. This shows that the major communities come from phylum Firmicutes. From this phylum, it has been 100% identified that the class of bacteria is Bacilli, the order of bacteria was Bacillales, the family was Bacillaceae, and the genus of bacteria was 100% Bacillus, as in Figure 1.

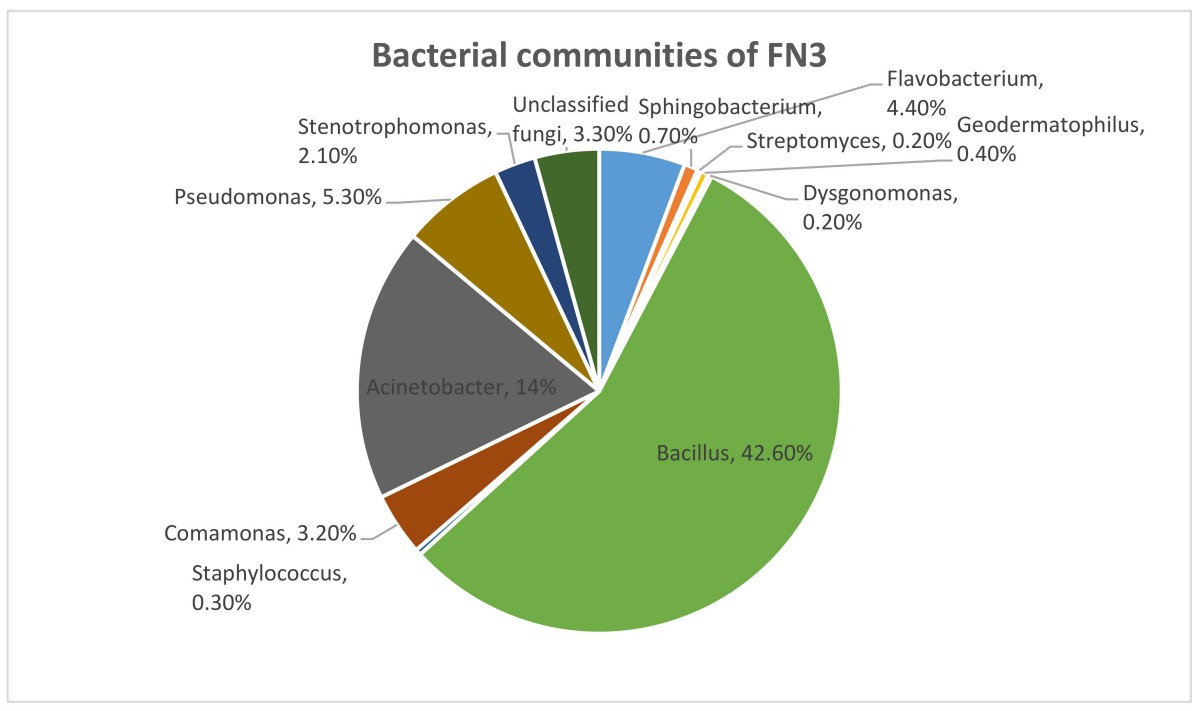

**Figure 1.** The bacterial communities that were identified in mixed bacterial culture FN3 by metagenomic analysis.

In Figure 2, it can be seen that a total of 115,952 reads of sequences composed of 97% unclassified fungi and 3% fungi from phylum Ascomycota were identified. In the phylum Ascomycota, the fungi genus identified was Candida. For an OTU to be categorized as "unclassified," based on the UNITE database, there is no "species" that is annotated up to a rank. This has given the perspective that it might be a novel species [21].

### 3.3. Optimisation of Significant Variables Using the Box-Behnken Design (BBD)

This study focused on the combined effects of four significant variables for the decolorization of the MY dye by mixed bacteria. To optimize the process variables for maximal dye decolorization, 29 experimental runs were conducted. Table 3 shows the experimental and response results together with the response predicted by RSM. Maximal and minimal dye decolorization was observed in runs 1 and 3, respectively, as shown in Table 3.

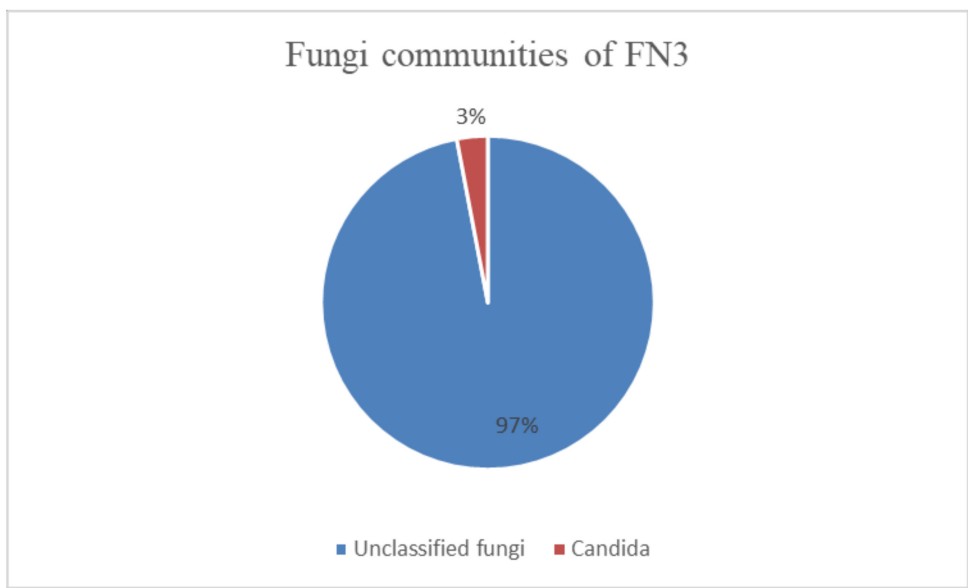

**Figure 2.** The fungi communities that were identified in mixed bacterial culture FN3 by metagenomic analysis.

**Table 3.** The BBD for the four independent variables on dye decolorization in actual and prediction values.

| Run | Factor 1 A: Dye Concentration (mg/L) | Factor 2: B: Glucose Concentration (%) | Factor 3: C: Ammonium sulphate Concentration (g/L) | Factor 4: D: pH | Decolorization (%) | Prediction by RSM (%) |
|-----|------|------|------|------|------|------|
| 1 | 50 | 2.00 | 0.55 | 6.75 | 88.35 | 95.8 |
| 2 | 50 | 1.25 | 0.55 | 7.50 | 87.14 | 75.53 |
| 3 | 125 | 1.25 | 1.00 | 6.00 | 5.85 | −4.49 |
| 4 | 125 | 1.25 | 1.00 | 7.50 | 45.43 | 40.95 |
| 5 | 200 | 1.25 | 0.55 | 7.50 | 38.00 | 27.87 |
| 6 | 200 | 1.25 | 1.00 | 6.75 | 17.85 | 20.95 |
| 7 | 200 | 1.25 | 0.55 | 6.00 | 2.86 | 13.62 |
| 8 | 125 | 2.00 | 1.00 | 6.75 | 29.92 | 40.61 |
| 9 | 50 | 1.25 | 0.55 | 6.00 | 23.00 | 32.27 |
| 10 | 50 | 0.50 | 0.55 | 6.75 | 14.20 | 22.57 |
| 11 | 125 | 2.00 | 0.55 | 6.00 | 39.00 | 27.30 |
| 12 | 125 | 1.25 | 0.10 | 6.00 | 0.00 | 10.34 |
| 13 | 200 | 1.25 | 0.10 | 6.75 | 9.60 | 9.97 |
| 14 | 125 | 0.50 | 1.00 | 6.75 | 0.00 | 6.41 |
| 15 | 125 | 0.50 | 0.55 | 6.00 | 8.02 | −0.32 |
| 16 | 125 | 0.50 | 0.10 | 6.75 | 0.93 | −10.61 |
| 17 | 125 | 1.25 | 0.55 | 6.75 | 61.08 | 58.77 |
| 18 | 50 | 1.25 | 1.00 | 6.75 | 50.35 | 44.98 |
| 19 | 200 | 0.50 | 0.55 | 6.75 | 14.86 | 13.27 |
| 20 | 125 | 1.25 | 0.55 | 6.75 | 48.02 | 58.77 |
| 21 | 50 | 1.25 | 0.10 | 6.75 | 60.36 | 52.25 |
| 22 | 125 | 2.00 | 0.10 | 6.75 | 61.19 | 53.98 |
| 23 | 125 | 2.00 | 0.55 | 7.50 | 74.48 | 77.81 |
| 24 | 200 | 2.00 | 0.55 | 6.75 | 41.29 | 38.78 |
| 25 | 125 | 1.25 | 0.55 | 6.75 | 60.50 | 58.77 |
| 26 | 125 | 1.25 | 0.55 | 6.75 | 63.77 | 58.77 |
| 27 | 125 | 1.25 | 0.55 | 6.75 | 60.49 | 58.77 |
| 28 | 125 | 1.25 | 0.10 | 7.50 | 6.20 | 22.41 |
| 29 | 125 | 0.50 | 0.55 | 7.50 | 0.00 | 6.69 |

ANOVA is a measurable investigation that is part of the analysis in RSM. It has been applied to identify the contrast between at least two groups that change in an experiment, and it is typically used to show that there is a significant outcome from the experiment. Along these lines, ANOVA was utilized to evaluate the significance of the model compared with the experimental values [22]. Table 3 shows the ANOVA of the regression parameters of the predicted response surface quadratic model for dye decolorization. The regression model is as follows:

$$\text{Dye decolorization} = 58.77 - 16.58 * A + 24.69 * B + 0.93 * C + 14.38 * D - 11.93 * AB + 4.57 * AC - 7.25 * AD - 7.59 * BC + 10.88 * BD + 8.34 * CD - 3.36 * A^2 - 12.81 * B^2 - 23.38 * C^2 - 18.09 * D^2$$

The predicted response fitted well with that of the experimentally obtained response. The adequate approximation of the selected model was measured by applying the diagnostic plots available in the Design Expert version 6.0.10 software, namely the externally studentizedresiduals plotted against the normal probability, the predicted versus studentizedresiduals, the runs versus studentizedresiduals, and the actual responses versus the predicted response values. Figure 3a shows that the externally studentized residuals plotted against the normal probability yielded a straight line showing the normal distribution of the experimental data. As shown in Figure 3b, the predicted versus externally studentized residual runs versus the externally studentized residuals and the actual responses versus the predicted responses, respectively, lie below the interval of $\pm 4.00$, indicating that the approximation of the model was good, with no data errors. Figure 3d illustrates the actual responses plotted against the predicted responses, which fit each other with correlation coefficients ($R^2$ and $R^2$ adj) of 0.913 and 0.825, respectively, for dye decolorization. Therefore, the developed model was suitable for predicting the efficiency of dye decolorization under the investigated conditions [23]. Table 4 shows that the F value of the model was 10.43 with a low probability value (F < 0.001), indicating that the model was significant for dye decolorization. On the other hand, a value of P that is less than 0.0001 is statistically significant for the quadratic equation of the model [24]. Values of p > F less than 0.0500 indicated that the model terms were significant, while values greater than 0.1000 indicated that the model terms were not significant. The lack of fit for the F test (0.0750) was statistically insignificant, implying that the model fitted the data. The non-significant value of lack of fit (>0.05) revealed that the quadratic model was statistically significant for the response and, therefore, it could be used for further analysis. The goodness of fit of the model was evaluated using the determination coefficient ($R^2$). In this case, the value of $R^2$ was 0.913 and the value of the adjusted $R^2$ was 0.825, which was in agreement with the predicted $R^2$ (0.525), indicating that the model was adequate for predicting the dye decolorization with any combination of the variables. An $R^2$ value close to 1.00 showed that the model was sufficiently strong in its prediction [25].

### 3.4. Determination and Validation of Optimal Conditions

The maximal decolorization was accomplished by the desirability function technique. This technique incorporates the wants and needs for every one of the factors to construct a system for deciding the connection between the anticipated color decolorization for every factor and the desirability of the reactions. The optimal conditions predicted by RSM were as follows: 73 mg/L of dye, 1.934% glucose, 0.433 g/L of ammonium sulfate, and pH of 7.907, which resulted in an overall 97.551% dye decolorization with a desirability value of 1 (Table 5). To verify this optimal condition, a validation experiment was performed according to the predicted condition obtained. The experimental result was compared with the given predicted value by measuring the deviation between both values. Verification experiments performed under the predicted conditions indicated the validity and adequacy of the predicted models. The results obtained through the validation of the experiment indicate the suitability of the developed quadratic models, and it may be noted that these optimal values are valid within the specified range of process parameters.

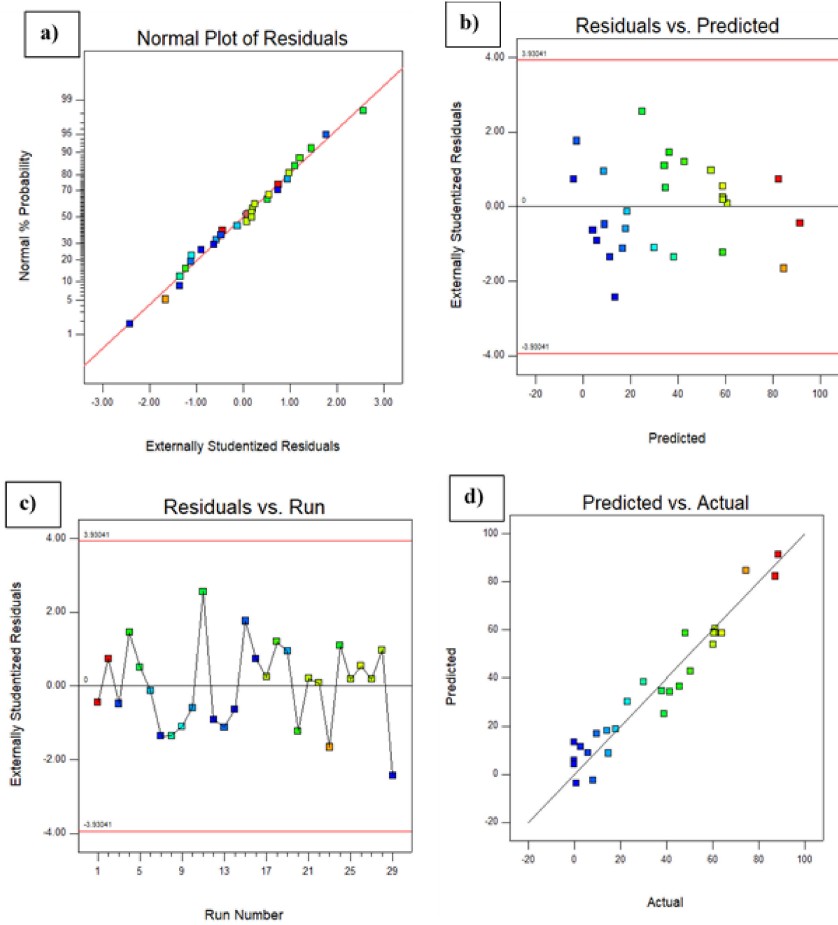

**Figure 3.** Diagnostic plots showing (**a**) the externally studentized residuals plotted against the normal probability, (**b**) the predicted versus the externally studentized residuals, (**c**) the run number versus the externally studentized residuals, and (**d**) the actual responses versus the predicted response.

**Table 4.** Analysis of variance (ANOVA) for the fitted quadratic polynomial for optimization of Metanil Yellow (MY) decolorization.

| Source | Sum of Squares | df | Mean Squares | F Value | *p*-Value Prob > F | |
| --- | --- | --- | --- | --- | --- | --- |
| Model | 20132.04 | 14 | 1438 | 10.43 | <0.0001 | significant |
| A-Dye concentration | 3298.09 | 1 | 3298.09 | 23.93 | 0.0002 | |
| B-Glucose concentration | 7312.19 | 1 | 7312.19 | 53.05 | <0.0001 | |
| C-Ammonium sulphate concentration | 10.3 | 1 | 10.3 | 0.075 | 0.7885 | |
| D-pH | 2480.26 | 1 | 2480.26 | 17.99 | 0.0008 | |
| AB | 569.3 | 1 | 569.3 | 4.13 | 0.0616 | |
| AC | 83.36 | 1 | 83.36 | 0.6 | 0.4497 | |
| AD | 210.25 | 1 | 210.25 | 1.53 | 0.2372 | |
| BC | 230.13 | 1 | 230.13 | 1.67 | 0.2173 | |
| BD | 473.06 | 1 | 473.06 | 3.43 | 0.0851 | |
| CD | 278.56 | 1 | 278.56 | 2.02 | 0.1771 | |
| $A^2$ | 73.09 | 1 | 73.09 | 0.53 | 0.4785 | |
| $B^2$ | 1064.3 | 1 | 1064.3 | 7.72 | 0.0148 | |
| $C^2$ | 3545.47 | 1 | 3545.47 | 25.72 | 0.0002 | |
| $D^2$ | 2123.12 | 1 | 2123.12 | 15.4 | 0.0015 | |
| Residual | 1929.84 | 14 | 137.85 | | | |
| Lack of fit | 1777.99 | 10 | 177.8 | 4.68 | 0.075 | Not significant |
| Pure error | 151.85 | 4 | 37.96 | | | |
| Cor total | 22061.88 | 28 | | | | |

**Table 5.** The optimum conditions were obtained using the desirability function technique.

| Dye Concentration (mg/L) | Glucose Concentration (%) | Ammonium Sulphate Concentration (g/L) | pH | Decolorization (%) | Desirability |
|---|---|---|---|---|---|
| 73 | 1.934 | 0.433 | 7.097 | 97.551 | 1 |

### 3.5. Effects of Different Initial Dye Concentration on Dye Decolorisation and Bacterial Growth

The mixed bacterial culture of FN3 was grown in optimized MSM with different MY dye concentrations ranging from 30 mg/L to 300 mg/L, as shown in Figure 4. From the graph, it is observed that a low dye concentration was decolorized at a higher rate compared to the high dye concentration by the mixed culture FN3.

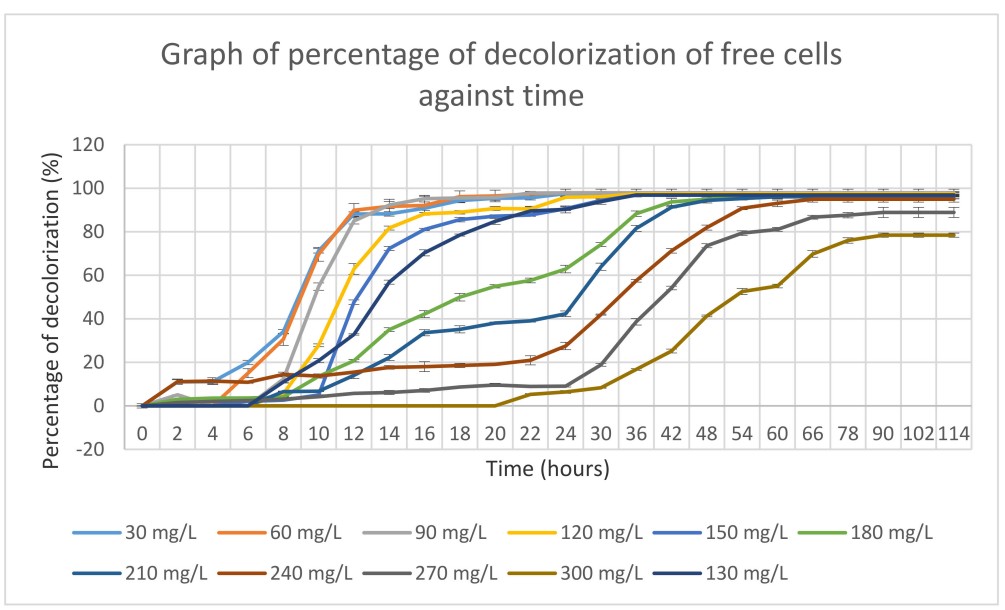

**Figure 4.** Dye decolorization percentages of free cells against time.

Meanwhile, mixed bacterial culture FN3 decolorized 300 mg/L of MY dye with an almost 80% decolorization rate for 114 h. Mixed bacterial culture FN3 fully decolorized MSM supplemented with 30–240 mg/L even though it took up to 66 h to decolorize. Meanwhile, for MSM supplemented with a dye concentration of 270 mg/L and 300 mg/L, mixed bacterial culture FN3 able to decolorize up to 80% dye decolorization only.

The dye decolorizdecolorization percentage by the mixed bacterial culture FN3 became static after a certain incubation time, which might be due to the inhibition of the dye by the mixed bacterial culture FN3.

### 3.6. Response Surface Plots of the Affecting Parameters

The surface response of the quadratic models was applied to visualize the effects of each experimental parameter, with two parameters maintained at the optimal value and the other two varying within the experimental ranges, as depicted in Figure 5. The three-dimensional response surface plots are the graphical representations of the regression equation. The main goal of the response surface is to efficiently track the optimum values of the variables such that the response is maximized. By analyzing the plots, the best response range can be calculated. Figure 5a shows the 3D surface response of the interaction effect of the glucose concentration and dye concentration. The dye decolorization increased when the glucose concentration was high and the dye concentration was low. This indicates that with an increasing supply of glucose, the bacteria used the glucose as a substrate for growth and, thus, increased the dye decolorization. Figure 5b shows the 3D surface response

of the interaction effect of the ammonium sulfate concentration and dye concentration. The dye decolorization increased when the ammonium sulfate concentration was between 0.2 and 0.5 g/L. Figure 5c shows the interaction effect of pH and dye concentration. The dye decolorization increased when the pH was between 6.0 and 7.5. The highest dye decolorization was achieved when the pH was around 7.5. This indicated that the bacteria preferred more alkaline conditions compared to acidic conditions for growth. For Figure 5d, the dye decolorization increased when the ammonium sulfate concentration was between 0.1 and 0.6 g/L, while the higher the glucose concentration, the higher the dye decolorization. It can be concluded that the bacteria need a nitrogen source also besides carbon source for decolorization of the dye. For Figure 5e, the higher the pH and glucose concentration, the higher the dye decolorization percentage. For Figure 5f, the dye decolorization increased when the pH was between 6.0 and 6.9 [26].

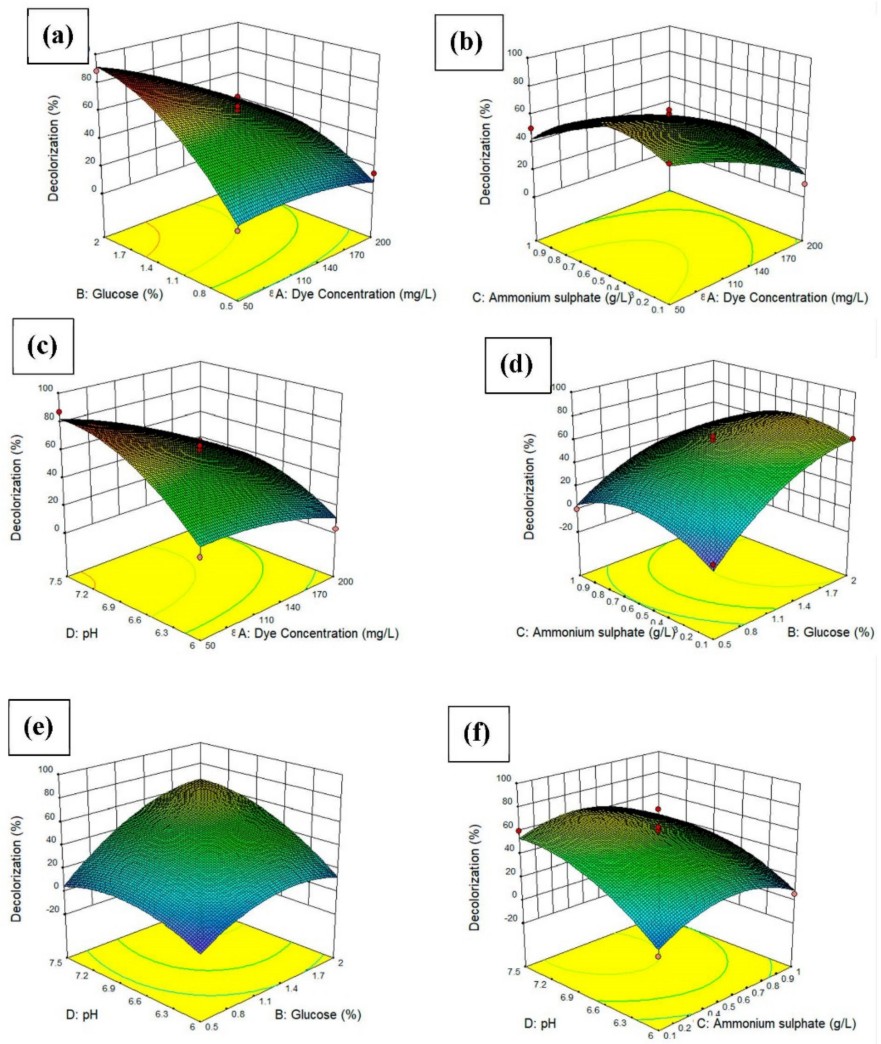

**Figure 5.** 3D plot and contour plot showing (**a**) the effects of dye concentration and glucose concentration on the percentage of dye decolorization, (**b**) the effects of dye concentration and ammonium sulfate on the percentage of dye decolorization, (**c**) the effects of dye concentration and pH on the percentage of dye decolorization, (**d**) the effects of glucose concentration and ammonium sulfate on the percentage of dye decolorization, (**e**) the effects of glucose concentration and pH on the percentage of dye decolorization, the (**f**) the effects of ammonium sulfate and pH on the percentage of dye decolorization.3.7 Effects of heavy metal ions on dye decolorization.

Heavy metals exist in various contaminated sites. Bioremediation of certain pollutants is often affected by the presence of heavy metals. Thus, studies to determine the effects of metal ions on bacterial bioremediation were performed. In this study, 1 mg/L of heavy metals, such as nickel (Ni), copper (Cu), lead (Pb), chromium (Cr), silver (Ag), zinc (Zn), mercury (Hg), and arsenic (As), were used.

Figure 6 demonstrates the percentage of dye decolorization of free cells with the presence of the heavy metals listed, and MSM with no addition of heavy metals served as the control. It was observed that the MY dye was decolorized in 24 h with the presence of heavy metals. This proved that the presence of heavy metals does not affect the dye decolorization. The percentage of dye decolorization (96.62%) was the highest with the addition of lead (Pb), whereas the lowest dye decolorization (56.19%) was achieved with the addition of copper (Cu).

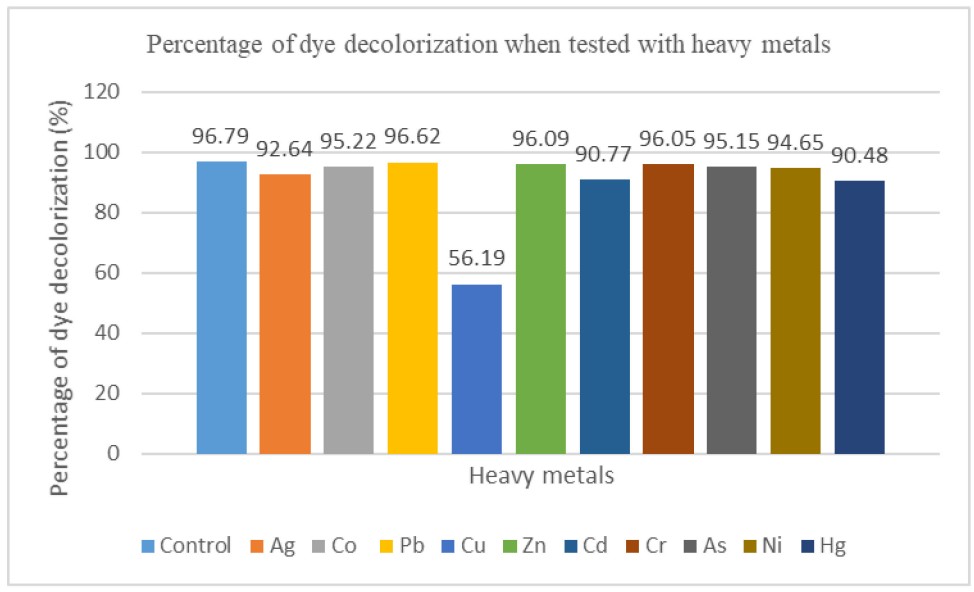

**Figure 6.** Bar chart showing the percentage of dye decolorization when tested with heavy metals.

## 4. Discussion

In this study, MY, a type of azo dye, was successfully decolorized within 24 h by a locally isolated mixed bacterial culture from Malaysian agricultural soil, namely FN3. According to a previous study, most of the microbial species that have been found to decolorize azo dyes originate from dye-contaminated wastewater or soil [27]. However, isolate FN3 has been isolated from agricultural soil. This indicates that dye-decolorizing bacteria can also be found in soil that has not been contaminated with dyes. Furthermore, in this work, a mixed bacterial culture was isolated instead of a pure culture. The advantages of mixed culture in dye decolorization have been reported in several reports [12,27,28]. The use of a mixed bacterial culture can result in a higher rate of dye decolorizdecolorization compared to the pure culture. From the metagenomics analysis of the mixed bacterial culture FN3, it was discovered that about 42.6% of the microbial community of FN3 is *Bacillus* sp. *Bacillus* sp. has been reported to decolorize azo dyes successfully [12,29]. It has been found that *Bacillus* sp. is well known and documented for its dye-decolorizing potential in previous studies [30]. About 42.6% of the microbial community was comprised of *Bacillus* sp., while another main species identified was *Acinetobacter* sp. (14%). A previous study reported the MY dye decolorization by the *Bacillus* sp. strain AK1 and *Lysinibacillus* sp. strain AK2 [2].

From the ITS diversity analysis, it was found that 97% of the mixed bacterial culture FN3 is "unclassified fungi." Based on the UNITE database, which is the reference database

in this metagenomics analysis of fungi, there is no "species" that is annotated up to a rank. This is indicated as "unclassified" fungi.

MY dye decolorization involves the enzymatic breaking of an azo bond (–N=N–) by the azoreductase enzyme. This process in azo dye decolorization eventually proceeds in two stages. In the first stage, the azoreductase breaks the azo bond, and this will produce colorless aromatic amines. The second stage involves the degradation of the colorless aromatic amines. In this study, the FN3 mixed bacterial culture needs aeration for growth. The mixed bacterial culture could not survive and could not decolorize MY with no aeration. It can be concluded that the mixed bacterial culture uses an aerobic mechanism. For the aerobic mechanism in the degradation of aromatic amines, it involves the replacement of the functional groups of the aromatic rings with other groups, such as hydroxyl groups. Next, two oxygen atoms are incorporated into the aromatic rings [9].

This study also indicates the potential use of the statistical optimization tool using RSM. It helps in finding the optimum values of each parameter and correlates each parameter to optimize the dye decolorization within a short period of time and for less cost. Before the use of RSM, scientists used OFAT to optimize certain conditions. This method is tedious and not cost-effective since a number of experiments are run and the parameters are not correlated. In this study, the dye decolorization was optimized and the optimum dye concentration was 73 mg/L with 97% decolorization. The chosen mixed bacterial culture needs glucose as the carbon source to grow and decolorize the dye. However, some bacteria are able to use the dye as the sole carbon source [31]. Even though our attempts to isolate the bacteria in order to use the dye as the sole carbon source have failed, the outsourcing of a carbon source such as glucose is being introduced to start the decolorization process. Another counted parameter is pH. Bacteria can live at various pH levels. In this study, the optimum pH suitable for the bacterial culture to decolorize the dye at an optimum level was 7.097, which is a neutral condition. From previous studies, *Bacillus* sp gives maximum dye decolorization at neutral pH [12].

The effect of different dye concentrations on the percentage of dye decolorization is significant in the study in order to identify the tolerance of mixed bacterial culture towards MY. The mixed bacterial culture FN3 was grown in MSM supplemented with 30 to 300 mg/L of MY. From this study, 240 mg/L of MY was completely decolorized. The concentration of 300 mg/L of MY was not decolorized completely by the mixed bacterial culture FN3. High concentrations of dyes cause the inhibition of metabolic processes of microorganisms. This is probably due to the toxic effects of the dyes on microorganisms through the inhibition of metabolic activity [32]. When the concentration of the dye is too high, the cells are saturated with the dye molecules, thus inactivating the transport system of the dyes [33]. The active sites of the enzymes that are responsible for decolorization may have been blocked by the dye molecules, thus reducing the decolorization efficiency [34].

Heavy metals are detrimental to microbes since they affect enzymatic functions, act as redox catalysts in the production of reactive oxygen species (ROS), disrupt ion regulation, and affect DNA and protein production [35]. Based on a previous study, the effects of heavy metal ions were tested on the dye decolorization. The dye decolorization was not affected by the presence of heavy metal ions [36,37].Dye decolorization is achieved with the addition of most heavy metals, such as Ag, Co, Pb, Zn, Cd, Cr, As, Ni, and Hg. This indicates that the mixed culture FN3 was able to tolerate the toxic effect to achieve decolorization. In this study, 56.19% of dye decolorization was achieved when 1 mg/L of Cu was added. The percentage of dye decolorization was low compared with the other tested heavy metals Thus, the low dye decolorization may be related to the inhibition of enzymes and metabolic pathways caused by Cu [33].

## 5. Conclusions

In conclusion, MY dye was decolorized by a mixed culture of bacteria and fungi from an agricultural soil sample. The mixed bacterial culture was successfully identified by 16s rRNA and ITS for bacterial and fungi metagenomic analysis, respectively. The result

showed that the highest percentage in the microbial communities was from the genus *Bacillus* and "unclassified" fungi. The MY dye decolorization by FN3 was also successfully optimized by RSM. The optimal conditions predicted by RSM were 73 mg/L of dye, 1.934% glucose, 0.433 g/L of ammonium sulfate, and a pH of 7.097, with decolorization up to 97.551%. The mixed cultures of FN3 also had high tolerance towards various heavy metals, as the dye decolorization of MY is not affected when heavy metals are added into the growth medium.

**Author Contributions:** Conceptualization, M.I.E.H., F.N.A.M.; data curation, M.I.E.H., F.N.A.M.; methodology, M.I.E.H.; supervision, M.I.E.H., S.B.A.W., S.S.A.G., U.H.Z., K.M., M.Y.A.S.; writing—Original draft, F.N.A.M.; writing—Review and editing, M.I.E.H., F.N.A.M. All authors have read and agreed to the published version of the manuscript.

**Funding:** This project was financed by funds from Putra Grant (GP-IPM/2017/9532800) and Yayasan Pak Rashid Grant UPM (6300893-10201).

**Conflicts of Interest:** The authors declare no conflict of interest.

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
