# Peer review of "Biostimulation of Microbial Communities from Malaysian Agricultural Soil for Detoxification of Metanil Yellow Dye; a Response Surface Methodological Approach"

_sustainability, doi:10.3390/su13010138_

Round 1

Reviewer 1 Report

Authors have taken a significant attempt to optimize conditions for Metanil Yellow (MY) dye decolorization. Obtained results are interesting and following minor modifications should be done before publishing.

  • First time authors have used RSM model to optimize the parameters. clearly indicate the advantages of using RMS model over other models. Also explain the novelty of the research over previous researches instead of the RMS model.
  • Figure 3 is not clear and must modify before publish
  • In Figure 4 legend has only 6 parameters, however there are lot of plots can be seen in the graph. clarify this one
  • In scientific research, at least 3 experiments should be carried out in-order to confirm the results, reproduce the graphs using 3 experimental results and include the error bars. 

Author Response

Refer attachment.

Reviewer 2 Report

Line 119: Please give the locations and coordinates for the collection sites from which these samples were retrieved - and describe them shortly (why were these chosen, have they been exposed to dyes etc). 

Line 68; note that aerobic appears twice, one of them should probably be replaced with anaerobic

Line 120: this method does not necessarily exclude fungi, did you observe any fungi? Note that some fungi species are very efficient in degrading different toxic compounds.

Line 215: replace kingdom with domain 

How was the DNA extracted and how was the sequencing procedure performed? You summarize some of ths in line 210-214, but this is in my opinion too superficial. There are many primers that can be used to sequence those genes, which ones did you choose? 

Did you submit your sequence data to an official sequence data base? - note, this is usually a request prior to publication of sequence data. 

Did you ever microscope your sample with the chosen enriched mixture - to see if it really only contained prokaryotes and fungi - it may after all also contain other eukaryotes like protists, nematodes - and these may then feed on the microbes and produce unknown consequences for your interpretations. 

Author Response

refer attachment

Reviewer 3 Report

I read with interest the manuscript “Biostimulation of Microbial Communities from Malaysia Agriculture Soil for Detoxification of Metanil Yellow Dye; Response Surface Methodological Approach "which describes the different phases of isolation, classification, and physiological characterization of a microbial communities actively degrading Metanil Yellow Dye.
The manuscript is interesting and the methodology utilized is correct. However, English is vary lacking, it must be carefully revised.

Here are some suggestions:

Lines 66-69: Cut the phrase “Decolorization under anaerobic condition means that the bacteria can only perform decolorization under the condition without the presence of oxygen. For anoxic condition, the bacteria can grow in both aerobic and aerobic condition but the decolorization can only occur under anaerobic conditions”. It is obvious.

Line 85: “According to the previous study” which study?

Lines 191-192: You should also add the concentrations of the single heavy metals.

Figure 1: You cannot mix in the same pie chart classes, orders, and families. Besides, the different colors do not allow to distinguish among the different bacteria. I suggest splitting the pie chart and writing close to the percentage the name of the genus.

Lines 238-239. Cut the phrase, it is a repetition.

Figure 3. The writings on the graphics are too small, it is almost impossible to read them.

Figure 4. The fives lines to the right of the figure have no explanation.

Lines 362-363. Cut the phrase “A mixed bacterial culture is a microbial community that consists of many species living together while pure culture of bacteria consists of one species of bacteria.”

Line 415-416. In your work the role played by mushrooms is not defined. They are low in species compared to bacteria, but their biomass has not been measured. Could the fact that the community is inhibited by Cu mean that fungi play a major role in degradation?

Line 428: Author contribution. Please specify the contribution of each author.

Author Response

Refer attachment

Round 2

Reviewer 2 Report

The authors have made a few changes of their original manuscript, but unfortunately, the English language and style is still of a very low quality. There is no way that this manuscript, irrespective of its content and potential great interest to readers, can be published. Pleases contact a native English speaking person or a professionally educated English language editor to improve your manuscript. After this I will be prepared to comment the scientific content of the manuscript. 

Author Response

The manuscript has been proofread, attached is proof of certificate
